# Exploiting the Molecular Properties of Fibrinogen to Control Bleeding Following Vascular Injury

**DOI:** 10.3390/ijms26031336

**Published:** 2025-02-05

**Authors:** Tanjot Singh, Muhammad Hasan, Thembaninkosi G. Gaule, Ramzi A. Ajjan

**Affiliations:** 1Leeds Institute of Cardiovascular and Metabolic Medicine, School of Medicine, University of Leeds, Woodhouse, Leeds LS2 9JT, UK; um22t2s@leeds.ac.uk (T.S.); t.g.gaule@leeds.ac.uk (T.G.G.); 2St James’s University Hospital, Beckett St, Harehills, Leeds LS9 7TF, UK; muhammad.hasan16@nhs.net

**Keywords:** fibrinogen, haemostasis, bleeding modulation, trauma, fibrin stabilisation

## Abstract

The plasma protein fibrinogen is critical for haemostasis and wound healing, serving as the structural foundation of the blood clot. Through a complex interaction between coagulation factors, the soluble plasma fibrinogen is converted to insoluble fibrin networks, which form the skeleton of the blood clot, an essential step to limit blood loss after vascular trauma. This review examines the molecular mechanisms by which fibrinogen modulates bleeding, focusing on its interactions with other proteins that maintain fibrin network stability and prevent premature breakdown. Moreover, we also cover the role of fibrinogen in ensuring clot stability through the physiological interaction with platelets. We address the therapeutic applications of fibrinogen across various clinical contexts, including trauma-induced coagulopathy, postpartum haemorrhage, and cardiac surgery. Importantly, a full understanding of protein function will allow the development of new therapeutics to limit blood loss following vascular trauma, which remains a key cause of mortality worldwide. While current management strategies help with blood loss following vascular injury, they are far from perfect and future research should prioritise refining fibrinogen replacement strategies and developing novel agents to stabilise the fibrin network. Exploiting fibrinogen’s molecular properties holds significant potential for improving outcomes in trauma care, surgical interventions and obstetric haemorrhage.

## 1. Introduction

Fibrinogen is a crucial plasma glycoprotein, notable for its high physiological plasma concentration of 2–4 g/L (6–13 μmol/L), making it the most abundant among the coagulation factors [1]. This significant concentration underlines its essential role as a key structural component in clot formation, where it plays a vital role in normal haemostasis and pathophysiological processes like tissue injury and inflammation [2]. These conditions can be triggered by vascular injury, whether due to trauma or surgical procedures, posing a significant risk of uncontrolled post-traumatic bleeding [3]. This remains the leading cause of potentially preventable mortality among trauma patients worldwide [4].

Following injury, the conversion of soluble fibrinogen into an insoluble fibrin network through the action of thrombin forms the structural foundation of the clotting mechanism [5]. However, coagulation defects associated with severe trauma can significantly impact fibrinogen function and consequent polymerisation, causing fibrinogen levels to reach a critical threshold that is unable to maintain normal haemostasis [6]. Although the European Society of Anaesthesiology’s guidelines recommend a minimum fibrinogen level of 1 g/L, even a small reduction in fibrinogen levels below 2 g/L can clinically increase the risk of perioperative and postoperative bleeding [6,7]. Understanding the underlying mechanisms and role of fibrinogen could facilitate the development of targeted therapies specifically addressing this molecule to more effectively reduce bleeding following vascular trauma. Currently, treatment approaches typically involve the use of antifibrinolytic agents, such as tranexamic acid (TXA) or, in more severe cases, coagulation proteins, including fibrin sealants [5,6], further detailed below.

This review aims to explore the molecular mechanisms by which fibrinogen modulates bleeding after vascular injury. By analysing recent advances in this field, we seek to provide a comprehensive understanding of how fibrinogen-related pathways can be harnessed therapeutically to enhance haemostasis and subsequent tissue repair, ultimately improving patient outcomes following vascular injury.

## 2. Fibrinogen Structure and Function

### 2.1. Subsection Description of Fibrinogen’s Molecular Structure: Aα, Bβ γ Chains

Fibrinogen is a 340 kDa soluble hexameric plasma glycoprotein, synthesised in the liver mainly by hepatocytes and encoded by three distinct genes (FGA, FGB, FGG) located within a 65 kb region on chromosome 4 [1]. The fibrinogen molecule is an elongated structure measuring approximately 45 nm, consisting of two outer D domains connected to a central E domain by coiled-coil regions (Figure 1a). Each subunit comprises three polypeptide chains, designated Aα, Bβ and γ, which are linked to the N-terminal region by five disulfide bridges [8,9,10]. The six fibrinogen polypeptides are arranged with their N-termini in a central E region and their C-termini radiating outward in a bilaterally symmetrical manner (Figure 1b). The C-terminal globular domains of the Bβ and γ chains are located in the D regions, while the Aα chain extends into an unstructured Aα region [11,12]. Once released, fibrinogen circulates in the blood at a high concentration with a half-life of approximately four days. Figure 1 provides a schematic representation of fibrinogen polypeptide chains.

### 2.2. Mechanism of Conversion from Fibrinogen to Fibrin by Thrombin

Injury to blood vessels triggers a haemostatic response. Exposure of the injured blood vessels initiates two processes that involve platelets and soluble plasma proteins. A collection of soluble plasma proteins acts together to initiate sequential activation of coagulation factors, leading to the formation of thrombin, as previously described [14]. Thrombin results in cleavage of part of fibrinogen (fibrinopeptide A and B) from the N-terminal regions of the Aα and Bβ chains [15], exposing polymerisation sites on the protein, the newly formed α- and β-‘knobs’ that insert into the corresponding ‘holes’ in the γC and βC regions of the D domain on a second fibrin monomer, facilitating the half-staggered alignment of fibrin monomers into protofibrils [16,17].

Blood flow (shear) during fibrin formation plays a pivotal role in regulating local thrombin generation by delivering procoagulant proteins and clearing activated enzymes [18]. This flow also aligns fibrin fibres, which significantly influences the formation of fibrin, as well as the clot’s mechanical integrity and resistance to fibrinolysis [19]. The presence of flow markedly alters the structural and mechanical properties of the fibrin network, increasing its density and stiffness while reducing fibre thickness and network porosity. These factors are particularly relevant when considering the physiological conditions that influence clot formation and stability in the context of vascular injury [20]. As the clot forms, the fibrinolytic system is activated to limit fibrin network extension. The zymogen plasminogen is converted to plasmin, by tissue plasminogen activator or urokinase plasminogen activator, which degrades the fibrin network [21].

### 2.3. Structural Role of Fibrin in Blood Clot Formation and Wound Healing

In addition to the role of fibrinogen in providing normal haemostasis and preventing blood loss, this molecule plays an important role in wound healing [22]. The fibrin matrix, which acts as a provisional extracellular matrix, is critical in tissue repair. This matrix facilitates the recruitment and adhesion of multiple cell types, including keratinocytes, fibroblasts, and leukocytes. These interactions promote cellular migration and contribute to wound healing by supporting tissue regeneration and remodelling [23].

## 3. Molecular Mechanisms of Fibrinogen in Bleeding Reduction

### 3.1. Cross-Linking Fibrin

Once fibrinogen is cleaved by thrombin into fibrin monomers, the newly exposed “knobs” on the Aα and Bβ chains bind to complementary sites (“holes”) in adjacent fibrin molecules, initiating protofibril formation. However, the resultant fibrin network gains functional resilience chiefly through Factor XIII-mediated cross-linking. Factor XIII (a transglutaminase of 325 kDa) circulates as a heterotetramer comprising two enzymatic A subunits and two non-enzymatic B subunits [24,25]. Thrombin cleaves and activates the A subunits in the presence of calcium ions (Ca^2+^), generating Factor XIIIa, which forges ε-(γ-glutamyl)lysyl bonds between fibrin strands [26].

Although the γ-chains constitute the classical cross-linking sites, Factor XIIIa also targets the flexible αC regions within the Aα chains [27]. These αC domains extend from the coiled-coil connectors and can dimerise or oligomerise, acting as “sticky ends” that promote lateral aggregation of fibrin protofibrils [28]. Cross-linking within and between αC regions enhances fibre cohesion, thereby reducing clot solubility and vulnerability to mechanical stress [29]. Furthermore, Ca^2+^ not only facilitates the conformational activation of Factor XIII but also contributes to the lateral packing of fibrin fibres, promoting robust three-dimensional architecture. Ultimately, this integrated cross-linking strategy yields a tightly woven lattice that minimises the risk of clot disruption under physiologically high shear conditions [30].

### 3.2. Platelet Aggregation and Clot Retraction: Molecular Interplay with Fibrin

#### 3.2.1. Fibrin Mechanisms of Platelet Aggregation

Alongside fibrin cross-linking, platelet aggregation is indispensable for stabilising the early haemostatic plug. Fibrinogen governs platelet bridging chiefly via the glycoprotein IIb/IIIa (GPIIb/IIIa) receptor on activated platelets [31]. Platelets interact with the subendothelial tissue following vascular injury through collagen and von Willebrand receptors, which, together with the exposure of these cells to soluble agonists (e.g., thrombin and ADP), trigger a cascade of signalling events that prompt GPIIb/IIIa to transition from a low- to a high-affinity state [32]. This exposes a binding interface specific for the C-terminal regions of fibrinogen’s γ-chains [33]. Each fibrinogen molecule can thus link two platelets concurrently, accelerating aggregate formation.

#### 3.2.2. Clot Retraction

Subsequently, the integrin–fibrinogen engagement initiates “outside-in” signalling, triggering cytoskeletal reorganisation. Platelet actin–myosin complexes contract, culminating in clot retraction [34]. By mechanically drawing fibrin fibres inward, retraction condenses the network, reducing pore size and clot permeability. This compact arrangement hinders the diffusion of fibrinolytic enzymes (e.g., plasmin) into deeper clot layers, augmenting clot longevity. Simultaneously, clot retraction narrows wound edges, aiding tissue approximation and creating a more favourable platform for cellular invasion and tissue repair [35].

### 3.3. Role of Fibrinolysis

#### 3.3.1. Activation of Plasmin and Fibrin Degradation

Haemostasis is maintained by a fine balance between clot formation and lysis to maintain physiological blood flow restoration of blood flow [36,37]. The activation of fibrinolysis is achieved by the conversion of plasminogen to plasmin, a serine protease that cleaves fibrin at specific lysine/Arginine sites [15]. Plasmin generation is catalysed by plasminogen activators, tissue plasminogen activator (tPA), and urokinase plasminogen activator (uPA) [38,39]. TPA-mediated plasminogen activation is fibrin-dependent, while uPA-mediated activation is fibrin-independent and less efficient.

The conversion of fibrinogen to fibrin exposes crucial activation sites for fibrinolysis that allow the binding of plasminogen and tPA to fibrin. Colocalization of plasminogen and tPA increases the efficiency of plasmin activation [40]. Once activated, initial plasmin action on fibrin results in the exposure of more plasmin sites, promoting fibrinolysis. Partial degradation products expose more tPA and plasminogen binding sites, initiating a feed-forward mechanism that accelerates clot lysis [41].

#### 3.3.2. Anti-Fibrinolytic Mechanisms

Fibrinolysis is spatially and temporally regulated by multiple inhibitory processes that prevent the premature lysis of the clot [15]. These inhibitory processes involve direct inhibition of serine proteases that drive fibrinolysis and indirect modulation of fibrinolysis. Antifibrinolytic safeguards ensure that the delicate equilibrium between clot formation and degradation is maintained, promoting the restoration of vascular integrity and blood flow [42].

Alpha-2 antiplasmin (α2AP) is a serine protease inhibitor (serpin) that primarily inhibits plasmin. Initially, α2AP interacts with plasmin non-covalently via its c—c-terminal lysine residues. Plasmin then cleaves a reactive centre loop on α2AP, resulting in an inactive covalent complex, irreversibly inhibiting plasmin.

During clot formation, α2AP is incorporated into the clot by covalent cross-linking to fibrin by Factor XIII. α2AP is cleaved by an α2AP cleaving enzyme, forming a truncated version, Asn^13^-α2AP, which is more efficiently incorporated into the clot. The incorporation of α2AP into the fibrin network, therefore, creates a localised shield against excessive plasmin-mediated proteolysis [43,44], preventing premature lysis and, thus, bleeding. A number of studies have shown that α2AP cross-linking to fibrin is a major determinant of clot resistance to fibrinolysis. Moreover, in vivo studies demonstrated that α2AP^-/-^deficient mice exhibited greater clot dissolution, further supporting the notion that α2AP is a major contributor to lysis resistance. α2AP regulates fibrinolysis by forming a complex with circulating plasmin and by creating a local shield against plasmin through Factor XIII-driven cross-linking to fibrin.

Like 2αAP, plasminogen activator inhibitor-1 and 2 (PAI-1 and PAI-2) are serpins that inhibit plasminogen activators, tPA and uPA [45]. Although both serpins inhibit tPA and uPA, PAI-1 is the major inhibitor, while PAI-2 is less abundant and is upregulated in conditions such as inflammation and pregnancy [46,47]. PAI-1 exists in two forms: active and latent, with the latter found complexed with vitronectin. PAI-1 has a high rate of association with tPA, which limits plasmin generation by confining fibrinolytic activity to sites where it is strictly required [45,48]. The active form has an exposed reactive loop that forms a covalent intermediate with tPA or uPA, thus inactivating them. Part of PAI-1 found at thrombus formation sites is platelet-derived, emphasising the interaction between the cellular and protein arms of coagulation to prevent early clot degradation. Therefore, insufficient PAI-1 activity can result in undue fibrin dissolution and an elevated bleeding tendency, as demonstrated in certain inherited or acquired PAI-1 deficiency states, although these conditions are relatively rare [49]. In contrast, high PAI-1 levels are associated with hypofibrinolysis and increased thrombosis risk, although evidence is lacking as to whether modulation of PAI-1 level and/or activity directly contribute to thrombotic or bleeding events [45].

Thrombin plays a central role in clot formation and stability. On initiating fibrin formation, thrombin activatable fibrinolysis inhibitor (TAFI) is converted to its active form TAFIa via the thrombin–thrombomodulin complex. TAFIa is a zinc-dependent metallocarboxypeptidase that down-regulates fibrinolysis by cleaving C-terminal lysine residues from fibrin [50]. The removal of the lysine residues stops plasminogen from binding to fibrin, thus reducing plasmin generation and limiting fibrinolysis. TAFIa regulates fibrinolysis in a threshold-dependent manner, where, at threshold levels premature, clot lysis is prevented by TAFIa inhibiting plasmin generation, which in turn prolongs clot lysis. However, when TAFIa concentrations fall below the threshold, the rate of fibrinolysis increases. In vivo studies suggest that TAFI deficiency per se has little effect on bleeding tendency but there are adverse effects for TAFI deficiency in the presence of other abnormalities in coagulation factors [51,52].

Additionally, α2-microglobulin is a broad-spectrum protease inhibitor that can bind plasmin and other proteolytic enzymes [53]. While it is not considered a primary antifibrinolytic mediator, its capacity to neutralise plasmin may provide an additional layer of control over fibrinolysis [54]. In certain pathological contexts, α2-microglobulin’s mechanism of protease inhibition can help stabilise clots and prevent excessive bleeding, though its overall impact on physiological haemostasis remains less clearly defined [46].

A summary of the procoagulant and anti-fibrinolytic proteins is provided in Table 1, while Figure 2 summarises the process of thrombus formation.

### 3.4. Clot Permeability and Contribution to Wound Healing

An additional layer of control lies in clot permeability, governed by fibre thickness, branching and overall density of the fibrin mesh [47]. Tightly packed fibres reduce pore size, restricting the migration of proteases like plasmin deeper into the clot matrix. Such structural arrangements, often influenced by Factor XIII cross-linking and platelet-driven compression, allow the haemostatic plug to endure sufficiently for tissue repair processes.

Concurrently, fibrin (ogen) provides a provisional extracellular matrix that shapes the dynamics of wound healing [22]. The dense fibrin scaffold not only traps circulating cells but also presents binding sites for leukocyte integrins, enabling swift immune cell recruitment. As the inflammatory phase subsides, fibroblasts and endothelial cells leverage this scaffold to migrate into the clot [55]. Over time, fibrinolysis and collagen deposition operate in tandem to replace the fibrin matrix with a more stable fibrous tissue. Therefore, from an integrative standpoint, fibrinogen ensures immediate bleeding control while providing a transient yet essential structure for vascular and tissue regeneration [56,57,58].

## 4. Clinical Applications of Fibrinogen

### 4.1. Trauma and Acute Bleeding

Severe trauma can precipitate a rapid depletion of fibrinogen, which often heralds the onset of acute traumatic coagulopathy (ATC). At a molecular level, several pathological processes converge to lower fibrinogen concentrations below the critical threshold required for effective clot formation [59]. One driver of this phenomenon is hyperfibrinolysis, wherein elevated levels of tPA convert plasminogen to plasmin, accelerating the proteolytic cleavage of fibrin. The degradation of fibrin produces fibrin degradation products (FDPs), which competitively interfere with fibrin polymerisation by binding to complementary sites on fibrin(ogen) monomers [60]. This not only disrupts nascent clot structures but can also feed back into further plasminogen activation, perpetuating a vicious cycle of fibrinogen consumption.

In parallel, the endothelium experiences extensive mechanical and inflammatory insults during trauma, leading to endothelial cell activation and the exposure of tissue factor (TF). Activation of TF initiates thrombin generation, which ordinarily cleaves fibrinogen to produce a robust fibrin matrix [26]. However, excessive thrombin production paradoxically heightens the consumption of fibrinogen and augments fibrinolysis, particularly if localised antithrombin mechanisms (e.g., thrombomodulin–protein C pathways) become dysregulated. Inflammatory cytokines, including interleukin-6 (IL-6), can stimulate acute-phase fibrinogen synthesis in hepatocytes yet simultaneously impair the molecular integrity of fibrinogen through oxidative modifications [11]. Oxidised fibrinogen polymers have compromised structural properties, rendering clots weaker and more vulnerable to plasmin-mediated degradation [61].

From a clinical standpoint, persistently low fibrinogen levels—commonly below 1.0–1.5 g/L—correlate strongly with ongoing haemorrhage and the need for transfusion [62]. In response, exogenous fibrinogen has emerged as a mainstay of trauma resuscitation, with two principal modalities in use. Cryoprecipitate is prepared from thawed fresh frozen plasma (FFP) and contains fibrinogen, Factor VIII, Factor XIII and von Willebrand factor (vWF). Despite being cost-effective, cryoprecipitate is associated with variability in fibrinogen content and potential infectious risks, owing to its derivation from pooled donors [63].

By contrast, fibrinogen concentrate is a purified product with a known fibrinogen content, enabling precise dosing. Fibrinogen concentrate rapidly restores fibrin polymerisation potential, fosters the formation of denser fibrin fibres resistant to proteolysis, and can be administered without blood type matching [64]. Yet, this approach is not without caveats. Fibrinogen concentrate is expensive and universal dosing protocols may not accommodate inter-individual variability in fibrinogen metabolism or the extent of haemorrhagic shock.

### 4.2. Obstetric Haemorrhage

Postpartum haemorrhage (PPH) constitutes a major global health concern, accounting for a significant proportion of maternal morbidity and mortality [65]. During pregnancy, plasma fibrinogen concentrations rise to 4–6 g/L, offering enhanced protection against bleeding at delivery [66]. However, under pathological conditions, such as uterine atony, placental retention, or obstetric lacerations, fibrinogen levels can plummet precipitously through robust fibrinolysis and sustained blood loss [67].

At a molecular level, the pathophysiology of PPH parallels that of acute traumatic coagulopathy in many respects. Excessive thrombin generation initially drives fibrin formation, yet hyperfibrinolysis mediated by tPA rapidly depletes fibrin(ogen) [60]. Moreover, oxidative modifications, together with local inflammatory mediators released from uterine tissue, can impair fibrin polymer assembly [11]. As fibrin(ogen) concentrations decrease below 2.0 g/L (or a FibTEM A5 < 12 mm on TEG), clot formation becomes increasingly ineffective [68]. Consequently, sustained haemorrhage ensues, jeopardising both maternal haemodynamics and organ perfusion.

Clinical approaches to restoring fibrinogen in PPH include cryoprecipitate and fibrinogen concentrate. Cryoprecipitate remains widely used in many regions due to its relative affordability; however, its variable fibrinogen yield and infection risk constitute significant drawbacks [63]. Fibrinogen concentrate, by contrast, contains a precisely measured quantity of fibrinogen, granting clinicians tighter control over the haemostatic response. However, the FIB-PPH trial investigating pre-emptive fibrinogen administration to reduce PPH did not show a benefit for this approach [69], although this therapy can be useful in those with low fibrinogen levels, emphasising the importance of tailored treatment with this protein [70]. Therefore, such an approach may be helpful in some patients, cost barriers impede universal adoption [71].

Measures such as uterotonics (e.g., oxytocin) and timely surgical interventions (e.g., uterine tamponade or arterial ligation) target the root causes of haemorrhage, including uterine atony and tissue retention [65]. Even so, the interplay of fibrinogen turnover, oxidative stress, and systemic inflammatory pathways underscores the complexity of managing PPH. In this context, TXA plays a crucial role as an adjunctive therapy. TXA is a lysine analogue that acts by competitively binding to the lysine-binding sites on plasminogen, thereby inhibiting its attachment to fibrin surfaces [72]. This blockade reduces plasmin generation and, in turn, curtails fibrin breakdown within the evolving haemostatic plug [73].

Future research should therefore clarify how best to integrate with fibrinogen replacement and uterotonics, enabling an individualised and multifaceted approach that safeguards maternal haemostasis without heightening thrombotic or neurological risks [74].

### 4.3. Cardiac and General Surgery

Cardiopulmonary bypass (CPB) during cardiac procedures exerts profound effects on haemostasis, often resulting in a complex coagulopathy characterised by reduced platelet count and function, hypothermia-induced enzyme dysfunction and above all, consumption and dilution of fibrinogen [26]. These perturbations are exacerbated by the non-physiological surfaces of the bypass circuit, which trigger platelet activation and cause shear stress on fibrinogen molecules [32]. Together, these factors undermine the formation of robust fibrin networks, culminating in excessive perioperative bleeding.

Fibrinogen supplementation seeks to reverse the depletion of fibrinogen and support the creation of structurally intact fibrin matrices that withstand the mechanical and enzymatic challenges inherent to CPB [75]. Fibrinogen concentrate is particularly effective in this respect because of its rapid onset and predictable fibrinogen dose, thereby minimising the use of additional allogeneic blood products. However, the cost of fibrinogen concentrate remains substantial and its large-scale use in prolonged or complicated procedures with extended CPB times is less established [76,77].

An alternative approach, cryoprecipitate, provides supplementary clotting factors, such as Factor VIII and von Willebrand factor (vWF), potentially benefiting patients with multifactorial coagulopathies. Yet cryoprecipitate’s fibrinogen content varies significantly between batches and its preparation delays can be ill-suited to acute intraoperative bleeding [63]. Hence, institutional preferences often oscillate between fibrinogen concentrate and cryoprecipitate, dictated by local cost structures, resource availability and the clinical urgency of bleeding control.

While addressing the molecular depletion of fibrinogen is paramount, successful outcomes in cardiac and major general surgeries also hinge on optimising platelet function, coagulation factor activity, and fibrinolytic control. Certain surgical settings require prophylactic antifibrinolytic agents—e.g., tranexamic acid—to counteract the heightened generation of plasmin [60]. Nonetheless, an overly robust antifibrinolytic strategy may theoretically predispose to thrombotic events, highlighting the necessity for a fine-tuned balance between procoagulant and fibrinolytic forces.

## 5. Therapeutic Implications and Future Research

There are a number of therapeutic modalities in current use for bleeding events, which are summarised in Table 2. Current and future therapeutic options are depicted in Figure 3.

### 5.1. The Role of TEG/ROTEM

Real-time viscoelastic haemostatic assays, rotational thromboelastometry (ROTEM), and thromboelastography (TEG) inform targeted fibrinogen replacement strategies [82,83,84]. These assays evaluate clot formation kinetics and strength, isolating the contribution of fibrin(ogen) to overall clot stability [85,86]. ROTEM and TEG capture the dynamic interplay of coagulation factors, platelets and fibrinolysis, providing near real-time assessments of global haemostasis [87].

There is growing evidence that supports the use of viscoelastic haemostatic point-of-care assays, such as TEG and ROTEM, for guiding fibrinogen replacement in obstetric haemorrhage [82,83,84]. TEG and ROTEM can expedite the detection of hypofibrinogenaemia, predict disease progression, and facilitate targeted fibrinogen supplementation in postpartum haemorrhage [87,88]. Although standard laboratory tests (e.g., Clauss fibrinogen) remain important, viscoelastic tests offer faster turnaround and a ‘current’ snapshot of haemostasis, which is particularly beneficial in acute or rapidly evolving PPH [88,89].

Quantra^®^ and Sonoclot^®^ have been introduced as alternative technologies for viscoelastic testing at point-of-care and critical care settings, analysing fibrin polymerisation, platelet function, and clot stability [90,91]. Preliminary studies demonstrate that these next-generation systems may enhance bedside coagulation monitoring in both obstetric and surgical settings by delivering simplified workflows and potentially reducing operator variability [92]. As a result, clinicians can make informed decisions about fibrinogen repletion—including the dose and timing—that align with ongoing haemostatic changes [93]. While further randomised studies are needed to confirm improvements in maternal outcomes, these point-of-care technologies hold promise for refining bleeding management protocols and minimising unnecessary transfusions. By enabling clinicians to tailor fibrinogen therapy to each patient’s molecular and functional profile, the likelihood of overtreatment or undertreatment diminishes. However, outcome studies on “precise dosing” are lacking and this remains an area for future research.

### 5.2. Fibrinogen Replacement

Current evidence, including large-scale investigations of fibrinogen replacement, supports early fibrinogen supplementation when fibrinogen levels fall below 1.5 g/L [94]. Such a proactive strategy mitigates the damaging effects of hyperfibrinolysis, normalises haemostasis, and reduces transfusion requirements for additional blood products [95]. Nonetheless, key challenges persist, including supply constraints, cost and logistical limitations, particularly in lower-resource settings where cryoprecipitate remains the only practical option. In addition, the interplay of diverse molecular pathways, such as the endotheliopathy of trauma and platelet dysfunction, must be addressed concurrently to optimise patient outcomes [96,97].

#### 5.2.1. Advantages of Fibrinogen Concentrates

Fibrinogen levels are recognised as a critical determinant of clot stability, with major bleeds often causing a swift drop in this essential coagulation factor. When haemorrhage requires surgical intervention, suboptimal fibrinogen concentration compounds bleeding risks, leading to deteriorating clinical outcomes [98]. At the molecular level, fibrinogen depletion undermines the formation of robust fibrin polymers, specifically reducing the density of inter-strand cross-links mediated by Factor XIIIa, and thus limiting the clot’s capacity to withstand haemodynamic stress. Furthermore, diluting coagulation factors with RBCs or crystalloids amplifies this deficit, as it disproportionately lowers fibrinogen relative to other clotting proteins [99].

In light of these mechanistic underpinnings, European guidelines recommend fibrinogen replacement therapy—particularly through FC—as a targeted intervention for acute blood loss [84]. Compared to FFP or cryoprecipitate, FC offers a more consistent and concentrated fibrinogen source. Cryoprecipitate, while containing fibrinogen alongside other factors (e.g., vWF, Factor XIII and Factor VIII), risks introducing proteolytic or prothrombotic elements in unpredictable quantities [77]. FFP administration likewise requires large volumes, increasing the likelihood of circulatory overload [100]. In contrast, FC delivers a higher fibrinogen-to-volume ratio, mitigating the dilutional impact on the fibrin mesh and limiting both haemodynamic strain and infection risk.

Clinical studies confirm these molecular advantages of FC over plasma-based therapies. Notably, the RETIC trial found that FC significantly enhanced fibrin polymerisation when compared with FFP, aligning with observations that stable fibrin matrices rely on an adequate supply of structurally intact fibrinogen [101]. Another randomised comparison suggested a mechanistic advantage: FC users experienced lower sepsis rates, suggesting that rapid fibrin restoration may also reduce inflammatory responses tied to prolonged bleeding [102]. In addition, analyses of FC versus cryoprecipitate confirm that the former consistently supplies higher fibrinogen content, which, at the molecular level, strengthens cross-linking and improves clot viscoelastic properties [103]. By contrast, cryoprecipitate’s unpredictable mixture of additional molecules (e.g., fibronectin, plasminogen) may inadvertently heighten the risk of either prothrombotic complications or suboptimal haemostasis [104].

#### 5.2.2. Observational Evidence in Postpartum Haemorrhage and Cardiac Surgery 

A key question was whether administering cryoprecipitate early rather than waiting for confirmed low fibrinogen levels might enhance outcomes in severe PPH. The pilot cluster randomised trial ACROBAT (Effect of early CRyoprecipitate transfusion versus standard care in women who develop severe postpartum haemorrhage) furnishes data on the feasibility of administering cryoprecipitate within ninety minutes of the first request for red blood cells [105]. Conducted across four large maternity units in London, this trial showed that prompt transfusion of cryoprecipitate is operationally viable in severe PPH and that an “early fibrinogen boost” could reduce overall transfusion requirements [106]. While ACROBAT was not designed to statistically confirm differences in morbidity or invasive interventions, it provided crucial pilot-level evidence that an early cryoprecipitate protocol is realistic and acceptable when supported by partial consent waivers and effective staff and patient communication. A larger, fully powered trial is now indicated to ascertain whether early fibrinogen correction results in fewer blood products used, lower surgical intervention rates, and improved maternal recovery.

Cryoprecipitate, however, remains at present a practical option in urgent scenarios, such as PPH. Data from Kamidani and co-workers demonstrated that early cryoprecipitate transfusion for women experiencing severe PPH could mitigate the risk of fluid overload by furnishing a relatively small but concentrated fibrinogen dose [107]. Although cryoprecipitate is less purified than fibrinogen concentrate and carries a theoretical infection risk, the study identified no clear viral or immunologic complications [107]. Nevertheless, more robust randomised evidence is needed to clarify whether early use confers sustained clinical advantages.

#### 5.2.3. Empirical and Observational Evidence from the FIBRES Trial and Post Hoc Analyses

Among recent RCTs, the Fibrinogen Replenishment in Surgery (FIBRES) study enrolled 735 adult cardiac surgical patients experiencing significant post-cardiopulmonary bypass bleeding, comparing fibrinogen concentrate with cryoprecipitate and stratifying patients by bypass duration (≤120, 121–180, and >180 min) [108]. While the primary endpoint-total allogeneic blood products transfused within 24 h—addressed immediate bleeding control, longer-term clinical outcomes, such as thrombotic events, were also assessed [108]. Importantly, fibrinogen concentrate remained non-inferior to cryoprecipitate, even when cardiopulmonary bypass exceeded three hours, countering the concern that cryoprecipitate’s additional factors (Factor VIII, Factor XIII, and von Willebrand factor) might improve haemostasis in protracted operations [109]. Further, there was no clear increase in thrombosis among recipients of fibrinogen concentrate, suggesting that targeted fibrinogen supplementation, guided by point-of-care coagulation tests, need not predispose patients to systemic hypercoagulability [108,109]. These results mitigate apprehensions that an “acute haemostatic benefit” might result in later “coagulation overload”. Instead, the FIBRES findings imply that fibrinogen concentrate remains both efficacious and safe beyond the immediate 24–48 h window. Going forward, meta-analyses that integrate findings from multiple RCTs-potentially including those involving mechanically assisted circulation or complex aortic surgery—will help confirm whether fibrinogen supplementation consistently avoids delayed thrombosis or similar late complications [109,110].

Beyond trials, such as FIBRES, numerous observational and retrospective investigations have enriched our understanding of fibrinogen therapy. In a long-term safety review, Rahe-Meyer and colleagues analysed 35 years of post-marketing data on RiaSTAP^®^/Haemocomplettan^®^ P combined with 52 additional clinical studies [77]. Despite substantial exposure over decades, the review found a low incidence of serious adverse drug reactions, including fatal thromboembolic events. Such rare events predominantly arose in patients who had other comorbidities or received concurrent procoagulants, implying that fibrinogen replacement per se rarely led to hypercoagulable states. Another salient finding was the absence of documented viral transmissions, reflecting robust pathogen inactivation during product manufacturing. These results position fibrinogen concentrates as a safer alternative to high-volume plasma transfusions—particularly in settings where cryoprecipitate or fresh frozen plasma can exacerbate infection or circulatory overload risks.

### 5.3. Tranexamic Acid Administration

#### 5.3.1. Trauma

Randomised trials, notably CRASH-2 and CRASH-3, have demonstrated the benefits of using TXA to maintain clot integrity in hyperfibrinolytic states. The CRASH-2 trial showed that TXA was most effective at reducing death due to bleeding when administered 1–3 h from injury. The study showed that when administered early, only 4.8% of the patients died in the TXA group compared to 6.8% of the placebo group (risk reduction 21%, *p* = 0.03). However, when TXA was administered late (after 3 h), a paradoxical increase in death due to bleeding was noted in the treatment group at 4.4% compared to 3.1% in the placebo group (risk increase of 44%, *p* = 0.004) [111]. Building on these findings, the CRASH-3 trial explored the effect of early administration of TXA in patients presenting with traumatic brain injury (TBI) and no major extracranial bleeding. The trial found that early administration of TXA (3 h from injury) resulted in a numerical 6% reduction in death in the whole group, which failed to reach statistical significance [112]. However, in a pre-specified analysis of individuals with mild to moderate head injury, a 22% reduction in death was demonstrated with the early use of TXA. Crucially, no rise in vascular occlusive events or seizures was observed. These data reinforce the principle that prompt inhibition of plasmin-mediated fibrinolysis can limit ongoing bleeding before coagulopathy worsens [113], while also highlighting the importance of rapid administration (ideally within 3 h) and careful patient selection to maximise survival benefits and minimise risks [113,114,115]. It can be argued, however, that the overall benefits are relatively small and more effective therapies are still required to further reduce the risk of mortality in such patients.

#### 5.3.2. Postpartum Haemorrhage

Although TXA is recommended alongside fibrinogen supplementation to bolster clot stability in active PPH [62], prophylactic administration of this agent remains a subject of debate. For instance, some studies have suggested that TXA might confer broad prophylactic benefits, while large-scale studies have shown nuanced results [116]. A multicentre RCT (TRAAP2) found that prophylactic TXA, administered at caesarean delivery alongside uterotonic agents, significantly reduced calculated blood loss exceeding 1000 mL or the need for red-cell transfusion by day two [73]. However, it did not reduce severe secondary haemorrhage-related clinical outcomes, highlighting the intricate, multifactorial nature of haemostatic regulation in obstetrics.

Beyond prophylaxis, active PPH management with TXA appears to have a more consistent benefit. In such scenarios, the agent is administered promptly upon diagnosing excessive bleeding, directly inhibiting plasmin-driven clot breakdown when fibrin (ogen) levels are at their most vulnerable [115]. Despite the demonstrated efficacy, dose optimisation in obstetrics continues to pose a challenge, as higher doses can potentially provoke neurological complications—ranging from headaches to rare but serious seizures [117]. Accordingly, clinicians increasingly personalise TXA usage by considering patient-specific factors such as baseline fibrinogen, inflammatory markers, and comorbidities (e.g., metabolic syndrome) [118].

### 5.4. Novel Strategies for Enhancing Fibrin Network Stability and Reducing Fibrinolysis

Hyperfibrinolysis remains a pivotal contributor to mortality in trauma-induced coagulopathy (TIC), arising from surges in plasminogen activators (tPA and uPA) that break down fibrin before secure clot formation can occur [119]. Recent viscoelastic haemostatic assays have permitted direct visualisation of clot dynamics, indicating that unchecked activation of plasmin destabilises fibrin polymers at multiple sites. This lytic process is exacerbated by diminished concentrations of endogenous inhibitors—most notably PAI-1 and α2-antiplasmin—thereby tipping the balance towards excess plasmin activity [120].

Several molecular mechanisms underlie this runaway fibrinolysis. Endothelial injury leads to the secretion of tPA, often magnified by catecholamines, adrenaline and other stress-related factors released systemically after trauma [121]. Meanwhile, reactive oxygen species generated by leukocytes and platelets can degrade PAI-1, removing an essential brake on fibrinolytic pathways [122]. The net result is a fibrin network depleted in both quantity and structural integrity, with insufficient cross-linking via Factor XIIIa to resist the onslaught of plasmin.

Efforts to thwart hyperfibrinolysis target plasmin or its upstream activators. Aprotinin, a protease inhibitor that blocks plasmin, kallikrein, and other serine proteases, exemplifies a molecularly oriented strategy to safeguard fibrin integrity [123]. By curtailing plasmin-mediated fibrin cleavage, aprotinin effectively reinforces the cross-linked network [124]. Similar approaches aim to fine-tune the activation state of Factor XIII, thereby intensifying fibrin cross-linking [125]. Moreover, combining antifibrinolytic drugs with fibrinogen supplementation can yield additive benefits: while FC compensates for lost fibrinogen, agents like tranexamic acid or aprotinin slow its enzymatic breakdown [110,126]. Through such dual molecular interventions, future therapeutic protocols may better preserve fibrin architecture, lessen the haemorrhagic burden, and ultimately improve survival across diverse clinical scenarios.

Despite these encouraging results, current approaches are not without drawbacks. For instance, high acquisition costs can restrict access to fibrinogen concentrate in resource-limited environments, while heterogeneity in patient responses may complicate standardised treatment protocols. In addition, off-label usage of certain haemostatic agents raises concerns about regulatory oversight, particularly if thromboembolic complications manifest. These unresolved issues highlight the need for further molecular research to develop refined strategies that enhance clot stability while minimising associated risks.

### 5.5. Personalised Medicine and Genetic Variations in Fibrinogen Genes

Although fibrinogen replacement appears to be beneficial across multiple clinical settings, growing evidence suggests that individual genetic differences in the fibrinogen alpha (FGA), beta (FGB), and gamma (FGG) genes can modulate plasma fibrinogen levels and responsiveness to antifibrinolytic agents, thereby potentially affecting clinical outcomes [127]. Findings from the AIRGENE study demonstrated that single nucleotide polymorphisms and haplotypes within FGA and FGB explained both inter- and intra-individual variability in fibrinogen levels among myocardial infarction survivors [128]. Polymorphisms, such as rs2070011 (FGA) and rs1800790 (FGB), emerged as significant modulators, particularly under conditions of elevated interleukin-6, which hints at an interplay between genetic makeup and inflammatory drivers of fibrinogen expression [127,128,129].

Further research, including Choi et al. on direct oral anticoagulants, has underscored the importance of specific FGG variants (for instance, rs1800792) in determining bleeding risk, implying that these genetic differences may similarly influence fibrinogen-targeted interventions, including tranexamic acid therapy [127]. Notably, this genetic dimension intersects with recent discoveries about the “antifibrinolytic gap” in severe injury, whereby tissue plasminogen activator is rapidly released before the endogenous antifibrinolytic response (via plasminogen activator inhibitors and α2-antiplasmin) fully develops [130]. Since some individuals may express more potent fibrinolysis or shutdown at baseline, genetic variations could further amplify these discrepancies, thus modulating the response to exogenous antifibrinolytics [131]. In principle, patients carrying haplotypes associated with upregulated fibrinogen expression or decreased plasminogen inactivation may derive heightened benefit from tranexamic acid or similar lysine analogues, whereas others might exhibit minimal fibrinolytic activity and be at greater risk of prothrombotic events if fibrinolysis is inhibited indiscriminately [127,128].

Although such pharmacogenomic tailoring remains largely aspirational, these studies affirm the need to integrate genetic testing with functional haemostatic measures in the future [127]. As data accumulate, personalising fibrinogen supplementation and antifibrinolytic use based on FGA, FGB, and FGG polymorphisms may become a realistic strategy for optimising haemostasis, minimising complications and, ultimately, improving patient-centred outcomes in major trauma, obstetric haemorrhage, and surgical procedures.

Additionally, from an economic and regulatory perspective, integrating personalised fibrinogen therapy into routine practice demands a careful cost–benefit analysis, especially when compared with the relatively lower-priced, widely available options of tranexamic acid, standard fibrinogen concentrates, or cryoprecipitate. For instance, TXA-used extensively in major haemorrhage settings-can cost as little as USD 2–USD 10 per dose, rendering it highly cost-effective in both trauma and obstetric contexts [81]. By contrast, fibrinogen concentrates vary widely in price-commonly USD 400–USD 800 per gram—and a typical adult dose might require several grams, depending on baseline fibrinogen levels [78]. Cryoprecipitate also has cost implications: a 10-unit pooled dose (commonly used in clinical settings for massive bleeding) can be USD 400–USD 700, but it provides less predictable fibrinogen content and carries additional logistical overhead (such as thawing, blood type screening, and infection risk) [78,79].

While personalised fibrinogen therapy—such as genotype-guided dosing or advanced point-of-care viscoelastic assays—may reduce overall transfusion requirements, it introduces other expenses. Genetic testing panels to identify FGA, FGB, and FGG polymorphisms can cost USD 200–USD 1000 or more per patient, depending on the platform and local reimbursement policies [132]. Likewise, real-time haemostatic monitoring devices (e.g., ROTEM or TEG) require not only capital investment for equipment but also specialised training for operators, thus inflating per-patient costs.

Moreover, regulatory hurdles are significant. To secure approval for a novel or genetically tailored fibrinogen therapy, developers must conduct extensive Phase I–III trials to confirm safety, efficacy, and consistent manufacturing standards. Such trials are typically protracted and expensive, often involving hundreds or thousands of patients in acute bleeding scenarios. This can slow the introduction of new interventions into the market, especially when competing with well-established, lower-cost agents like TXA and cryoprecipitate.

## 6. Future Directions

### 6.1. Stabilising the Fibrin Network with Affimers

One promising approach centres on Affimer proteins—engineered molecular scaffolds designed to bind specific epitopes with high affinity. By targeting critical sites within fibrin (ogen), Affimers can modulate fibrinolysis at the structural level [133]. In proof-of-concept studies, these small, non-antibody agents have been shown to delay plasmin generation by inhibiting plasminogen or tPA binding [134]. Mechanistically, Affimers appear to alter the fibrin mesh conformation, thereby restricting enzymatic access to cleavage sites. Unlike large-scale fibrin sealants, Affimers may minimise immunogenic risk and infection hazards. Additionally, their compact structure facilitates better diffusion and potentially more uniform distribution throughout the clot, offering a tailored means to preserve fibrin under high-shear or hyperfibrinolytic conditions [133,134].

Their smaller size and stability, compared to traditional antibodies, make them amenable to bacterial production, potentially streamlining batch consistency [133,135]. Furthermore, the technology has been extended to affinity purification of fibrinogen, demonstrating the high target specificity of this technology [136].

Despite this promise, Affimers remain at a relatively early translational stage. Rigorous Phase I–II trials are needed to confirm safety, efficacy, and immunogenic profiles. In addition, Good Manufacturing Practice (GMP) compliance poses a significant challenge, particularly if custom conjugations (e.g., PEGylation) are required to enhance pharmacokinetics [137]. Consequently, per-unit costs may prove to be higher than those of established haemostatic agents, which may limit broader adoption until economies of scale or new manufacturing platforms reduce production expenses [133].

### 6.2. Increasing Resistance to Lysis via α2-Antiplasmin

Another avenue aims to augment antifibrinolytic protein incorporation—particularly α2-antiplasmin (α2AP)—into the clot. α2AP normally becomes cross-linked to fibrin via Factor XIIIa, forming a potent barrier against plasmin activity. By engineered modifications that enhance α2AP’s fibrin-binding or by promoting more compact fibrin architectures, one can improve α2AP embedding [138]. Evidence suggests that denser fibrin networks incorporate higher concentrations of antifibrinolytic molecules, which slows lysis [139]. Hence, altering clot structure at the molecular level—for instance, through enhanced cross-linking or local upregulation of α2AP—could further amplify clot robustness in high-risk bleeding scenarios.

### 6.3. A Combined Molecular Approach

Given the intricacies of fibrin biology, a multifactorial intervention is likely to provide the greatest clinical benefit. Combining fibrinogen replacement to ensure robust protofibril formation, Affimers or similar binders to stabilise the fibrin matrix and antifibrinolytic therapies (e.g., tranexamic acid) to reduce clot breakdown, could yield a synergistic effect [140]. Such an integrated strategy targets diverse molecular events—ranging from upstream fibrin polymerisation to downstream plasmin inhibition—thus reinforcing clot strength without disproportionately elevating thrombotic risk [141]. Future clinical trials employing point-of-care coagulation monitoring could optimise dosing regimens, tailoring the ratio of fibrinogen replacement to Affimer administration and antifibrinolytics for individual patient profiles. In doing so, personalised medicine may advance a new era of haemostatic care, with lower mortality and morbidity rates linked to massive bleeding.

Overall, these future directions highlight the scope for innovation in haemostatic therapies, bridging existing limitations in cost, availability, and safety. By harnessing molecular mechanisms more precisely, clinicians can not only improve clot stability but also minimise adverse events, ultimately transforming clinical outcomes for patients facing severe bleeding challenges.

## 7. Summary of Key Findings

Fibrinogen sits at the nexus of the clotting cascade, its molecular integrity and abundance dictating whether haemostasis proceeds effectively or devolves into pathological bleeding. By facilitating protofibril formation, promoting Factor XIII-driven cross-linking, and contributing to the structural cohesion of fibrin networks, fibrinogen underpins the physical and functional robustness of blood clots. As elucidated in this review, deficiencies in fibrinogen-or its supporting molecular mechanisms-can exacerbate haemorrhagic risks across a range of clinical contexts, from traumatic coagulopathy to obstetric haemorrhage and cardiopulmonary bypass. 

In addressing these shortcomings, emerging interventions, such as fibrinogen concentrates, advanced viscoelastic monitoring, and novel antifibrinolytic strategies (including Affimers and engineered α2-antiplasmin), hold significant therapeutic promise. Indeed, optimising fibrinogen replacement protocols, reinforcing fibrin stability through targeted protein modulators, and mitigating excessive fibrinolysis via combination regimens could collectively usher in a new era of precision haemostasis. Such an integrated approach will require rigorous translational research to balance efficacy against potential thrombotic hazards, yet it stands poised to transform bleeding management by individualising therapy to a patient’s specific molecular and clinical profile. Continued exploration into the molecular nuances of fibrinogen function and its downstream pathways remains integral to refining haemostatic care and enhancing outcomes in diverse surgical and emergency settings.

## Figures and Tables

**Figure 1 ijms-26-01336-f001:**
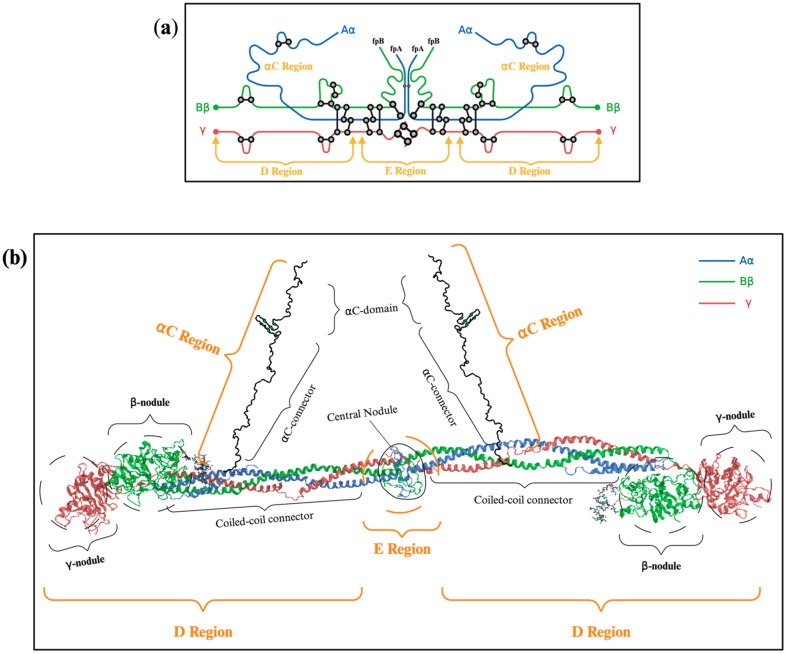
Schematic Representation of Fibrinogen Polypeptide Chains: (**a**) The schematic diagram illustrates the polypeptide chains of fibrinogen, where the Aα, Bβ and γ chains are depicted as lines with lengths corresponding to the number of amino acid residues in each chain. Various structural regions are labelled accordingly (adapted from Zhmurov et al. 2011 [13]); (**b**) This panel presents a model of the fibrinogen structure derived from the crystal structure of fibrinogen (PDB: 3GHG) and the NMR structure of the αC domain (PDB: 2BAF). The (Aα/Bβ/γ)_2_ hexamer configuration results in five distinct regions: the E region, two D regions, and two αC regions. The E region, located at the centre, comprises the N-terminal ends of all the chains (Aα in blue, Bβ in green, and γ in red). The D region includes a triple-stranded coiled-coil connector along with the β- and γ-nodules. The αC region consists of the Aα chain, which includes the αConnector and the αC domain.

**Figure 2 ijms-26-01336-f002:**
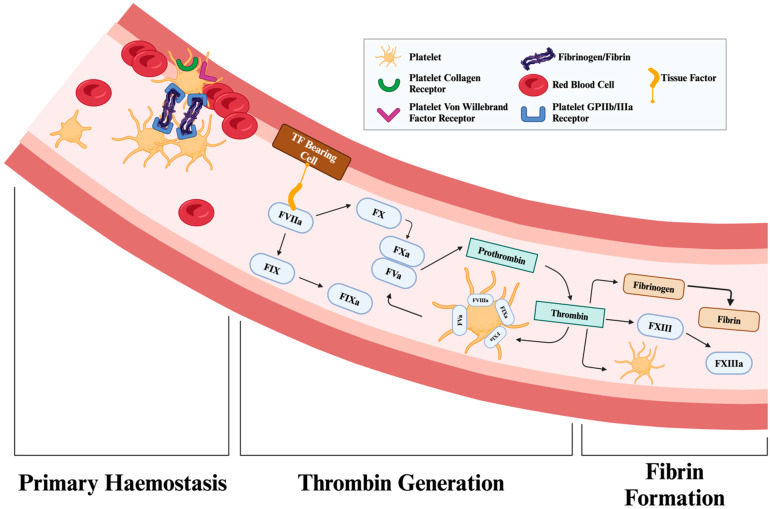
Thrombus formation. Primary haemostasis: Platelets adhere to exposed subendothelial collagen via von Willebrand factor and become activated, undergoing a shape change and aggregation through glycoprotein IIb/IIIa receptors, thus forming an initial platelet plug. Thrombin generation: A cascade of proteolytic activations culminates in the production of thrombin from prothrombin. Thrombin then cleaves fibrinopeptides A and B from fibrinogen, exposing sites (“knobs”) that bind to complementary “holes,” promoting the assembly of protofibrils. Fibrin formation and stabilisation: Soluble fibrin monomers polymerise into a fibrous lattice. Concurrently, Factor XIIIa cross-links fibrin strands, reinforcing the clot’s structural integrity under haemodynamic stress. This final step is essential for sustaining clot stability and limiting excessive fibrinolysis. The interaction between platelets and fibrin networks ensures the effective formation of a blood clot.

**Figure 3 ijms-26-01336-f003:**
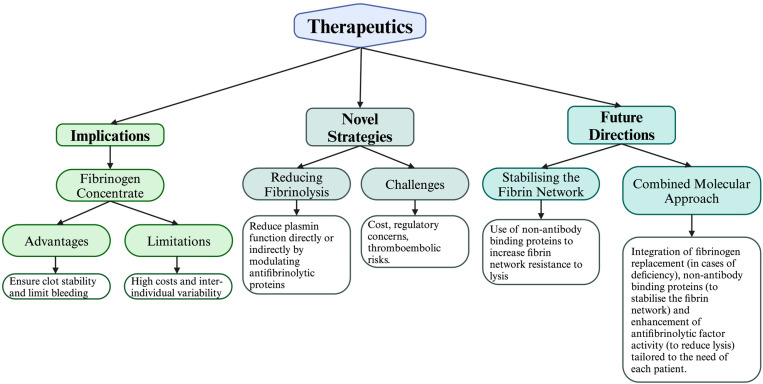
Schematic Representation of Therapeutic Implications, Novel Strategies, and Future Directions in Haemostatic Care. This figure summarises the multifactorial approaches to optimising fibrin stability and reducing fibrinolysis, divided into three major components: Therapeutic Implications: This section highlights the advantages of fibrinogen concentrates (FC), including their superior efficacy over FFP and cryoprecipitate in restoring clot stability and mitigating inflammatory risks. The limitations of FC, including high costs and variable responses, are also noted. Novel Strategies: Recent developments to enhance fibrin network integrity include the use of antifibrinolytics like tranexamic acid or aprotinin, as well as strategies targeting Factor XIIIa cross-linking to strengthen fibrin polymers. Future Directions: Emerging therapies, such as Affimers and α2-antiplasmin integration, are depicted as innovative molecular tools to stabilise the fibrin matrix. A combined approach integrating fibrinogen replacement, Affimers and antifibrinolytics, tailored by point-of-care monitoring, represents the potential for personalised haemostatic care with reduced thrombotic risks and improved outcomes.

**Table 1 ijms-26-01336-t001:** Procoagulants and Antifibrinolytic proteins involved in thrombosis.

Name	Class	Mechanism
Factor XIII	Protein transglutaminase	Activated by thrombin; catalyses bonds between g chains of fibrin molecules; crosslinks molecules to anti fibrinolytic molecules, e.g., α2-antiplasmin
GPIIb/IIIa	Integrin	One fibrinogen molecule links two receptors via g chain sequence, forms platelet aggregates
Calcium (Ca2+)	Ion (Ca^2+^)	Binds with varying levels of affinity to different fibrinogen chains, highest affinity on the γ1 chain, increase the rate of lateral aggregation between fibres of fibrinogen
α2AP	Serine protease inhibitor	Main physiological inhibitor of plasmin; Factor XIII cross-links it to fibrinogen
TAFI	Zinc-dependent metallocarboxypeptidase	Found in platelet α-granules, accumulates when platelets gather; cleaves off C-terminal lysine residues in fibrinogen, thus reducing binding of plasminogen
PAI-1	Serine protease inhibitor	The main inhibitor of both tPA and uPA, thereby modulating plasmin generation and controlling fibrin degradation to avert excessive fibrinolysis.
PAI-2	Serine protease inhibitor	Endogenous inhibitor of urokinase-type plasminogen activator; cross-linking to Lysine residues through glutamine residues on the PAI-2 chain
α_2_-Macroglobulin (α_2_M)	Broad-spectrum protease inhibitor	Binds and inactivates a range of proteases, including plasmin, through a conformational trap mechanism. By sequestering proteolytic enzymes, α_2_M adds an additional layer of protection against excessive fibrinolysis.

**Table 2 ijms-26-01336-t002:** Current Treatment Modalities for Bleeding Events, Their Mechanisms, Applications, Advantages, and Drawbacks.

Treatment Modality	Mechanism of Action	Applications	Advantages	Drawbacks	Cost
Fibrinogen concentrate	Directly replenishes fibrinogen levels for clot formation and stability	Trauma, obstetric haemorrhage, cardiac surgery	Rapid, precise dosing; predictable effects; low infection risk	High cost; limited availability in resource-limited settings	USD 400–USD 800 per gram [78]
Cryoprecipitate	Supplies fibrinogen along with Factor VIII, XIII and vWF	Trauma, obstetric haemorrhage, cardiac surgery	Widely available; cost-effective in many settings	Variable composition; freezing requirements; infection risk	USD 400–USD 700 [78,79]
Fresh Frozen Plasma (FFP)	Provides a broad range of clotting factors to address coagulopathy	Massive transfusion in trauma or surgery	Provides broad coagulation support; widely available	Requires large volumes; risk of transfusion reactions	USD 132–USD 477 [80]
Tranexamic Acid (TXA)	Inhibits plasminogen activation to reduce fibrinolysis	Trauma, obstetric haemorrhage, surgical bleeding	Simple administration; effective in reducing fibrinolysis	inconsistent action; efficacy depends on fibrinogen levels	USD 2–USD 10 (USD) per dose [81]

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
