# Peer review of "Exploiting the Molecular Properties of Fibrinogen to Control Bleeding Following Vascular Injury"

_ijms, 2025, doi:10.3390/ijms26031336_

Round 1
Reviewer 1 Report
Comments and Suggestions for Authors
Manuscript Singh et al. offer a comprehensive review of the structure, function, and key interactions of fibrinogen in the process of hemostasis, contributing to a better understanding of its clinical significance.
The authors clearly outline current and experimental therapeutic strategies, including fibrinogen concentrates, antifibrinolytic agents, and innovative molecular approaches such as Affimers
The text is well organized, and key findings and suggestions for future research are summarized, making it easy to follow the material
Although the work provides a rich theoretical basis, there is a lack of empirical evidence supporting the effectiveness of new therapeutic strategies in real clinical conditions. It is recommended to include more data from randomized controlled trials and meta-analyses that could confirm the effectiveness of the described methods. For example, data on the long-term outcomes of patients who received fibrinogen concentrates compared with standard treatment would help validate proposed therapies.
Although the paper mentions personalized medicine, it does not sufficiently analyze how genetic and physiological differences between patients can affect the effectiveness of treatment. Integration of pharmacogenetic studies is recommended to understand how different patients respond to fibrinogen therapy. For example, analysis of genetic variation in the FGA, FGB, and FGG genes could help predict patient response to fibrinogen concentrates and antifibrinolytic agents.
New therapeutic approaches, such as Affimers and personalized fibrinogen therapy, carry high development and implementation costs, which may limit their availability - Authors could look more closely at the cost-effectiveness of these treatments compared to existing options, as well as the regulatory hurdles that need to be overcome before they can be used. these drugs are widely used.
The work mainly focuses on the acute effects of fibrinogen and its analogs but does not sufficiently consider the potential long-term consequences, such as the risk of thrombosis or imbalance in the coagulation system in patients receiving fibrinogen supplements.
Comments on the Quality of English LanguageThe English could be improved to more clearly express the research
Reviewer 2 Report
Comments and Suggestions for Authors
I enjoyed reading this comprehensive review on the utility of fibrinogen to limit bleeding after vascular injury. I have some comments:
- little is mentioned about fibrinolysis, currently limited to lines 90-91 on page 2. Perhaps another section before section 3.3 (antifibrinolytic pathways)? Include tPA, uPA, plasminogen and plasmin and how they interact with fibrin(nogen) to effect fibrinolysis
- Table 1: PAI-1 is not mentioned. Alpha-2-macroglobulin is another major fibrinolytic inhibitor.
- Section 3.3: should be stressed that PAI-1 is the major inhibitor of tPA.
- Lines 162-164: 'Excessive PAI-1 activity can predispose to thrombosis' - high PAI-1 has been found in some studies in diabetes, metabolic syndrome and other thromboinflammatory states, however, whether it is a byproduct of inflammation/endothelial dysfunction or directly involved in thrombosis has not been conclusively proven. Need references included. Also 'insufficient activity may de-stabilise clots' - please provide references to support this.
- Needs a little more description about tranexamic acid - explain mechanism of action and current evidence in clinical practice. There are now RCTs in trauma, obstetrics and major surgery - suggest a brief review of the evidence in the relevant sections.
- line 278 - please include reference.
- Can thromboelastography also inform fibrinogen replacement in obstetric PPH and surgery? Are there other point-of-care assays under investigation to guide fibrinogen replacement?
Round 2
Reviewer 2 Report
Comments and Suggestions for Authors
The authors have comprehensively addressed the previously made comments and improved the manuscript.